# CH_3_NH_3_PbBr_3_ Thin Film Served as Guided-Wave Layer for Enhancing the Angular Sensitivity of Plasmon Biosensor

**DOI:** 10.3390/bios11110415

**Published:** 2021-10-23

**Authors:** Leiming Wu, Yuanjiang Xiang, Yuwen Qin

**Affiliations:** 1Advanced Institute of Photonics Technology, School of Information Engineering, Guangdong University of Technology, Guangzhou 510006, China; leiming_wu@gdut.edu.cn; 2Guangdong Provincial Key Laboratory of Information Photonics Technology, Guangdong University of Technology, Guangzhou 510006, China; 3School of Physics and Electronics, Hunan University, Changsha 410082, China

**Keywords:** guided-wave surface plasmon resonance, Au–perovskite hybrid structure, biosensor

## Abstract

CH_3_NH_3_PbBr_3_ perovskite thin film is used as a guided-wave layer and coated on the surface of an Au film to form the Au-perovskite hybrid structure. Using the hybrid structure, a perovskite-based guided-wave surface plasmon resonance (GWSPR) biosensor is proposed with high angular sensitivity. First, it is found that the electric field at the sensing interface is improved by the CH_3_NH_3_PbBr_3_ perovskite thin film, thereby enhancing the sensitivity. The result demonstrates that the angular sensitivity of the Au-perovskite-based GWSPR biosensor is as high as 278.5°/RIU, which is 110.2% higher than that of a conventional Au-based surface plasmon resonance (SPR) biosensor. Second, the selection of the coupling prism in the configuration of the GWSPR biosensor is also analyzed, and it indicates that a low refractive index (RI) prism can generate greater sensitivity. Therefore, the low-RI BK7 prism is served as the coupling prism for the proposed GWSPR biosensor. Finally, the proposed GWSPR sensing structure can not only be used for liquid sensing, but also for gas sensing, and it has also been demonstrated that the GWSPR gas sensor is 2.8 times more sensitive than the Au-based SPR gas sensor.

## 1. Introduction

Surface plasmon resonance (SPR) is an optical phenomenon formed by the interaction between the incident TM-polarized light and the free electrons on metal surface. Using the Kretschmann–Raether (KR) configuration [1], the SPR phenomenon [2,3,4] can be excited with a coupling prism. When a light wave is incident from an optically dense medium into an optically thinner medium, the attenuated total reflection (ATR) will occur if the incident angle is greater than the critical angle, and an evanescent wave (EW) will be formed along the critical interface. Metals such as Au, Ag, Al, and Cu have a large number of free electrons on their surface. These free electrons can easily form a surface plasma wave (SPW) that oscillates perpendicular to the metal surface under the guidance of incident TM-polarized light. When the frequency, amplitude and phase of the EW and SPW are well matched, they will couple and resonate, thus forming a resonance dip in the reflectivity curve, that is, the SPR signal [4]. The SPR signal is sensitive to the changes of the surrounding environment, and is widely used as a detection signal for real-time sensory monitoring of the biological interaction [5,6,7] or chemical reaction [8,9,10] process in the sensing medium. There are many kinds of biochemical reactions in the sensing medium layer, such as human immunoglobulin G (IgG) detection [11,12,13], single-stranded DNA/RNA sensing [14,15,16], and the detection of harmful chemical substances [17,18,19]. The progress of these RI reactions will cause changes in the refractive index of the surrounding environment. Therefore, the monitoring of these biological or chemical processes can be transformed into the detection of changes in the RI of the sensing layer. The conventional SPR biosensor, which is made of 50 nm thick Au film coupled with a prism, has good stability, but the sensitivity is not high enough. Previous reports indicate that the sensitivity of the conventional SPR biosensor based on the BK7 prism is 137°/RIU [20], which results in it being difficult to detect more slight RI changes of sensing medium. Hence, it is necessary to explore other SPR biosensors with higher sensitivity.

In order to enhance the sensitivity of the SPR sensor, previous researchers have performed many investigations and proposed various novel sensor structures, such as long-range SPR (LRSPR) [21,22,23] and guided-wave surface plasmon resonance (GWSPR) [4,24,25]. In optical sensors, the sensitivity measurement methods are divided into four categories [4], including angular sensitivity, intensity sensitivity, spectral sensitivity, and phase sensitivity. For LRSPR sensors, the resonance dip is relatively narrow, which is suitable for measurement with intensity sensitivity. As for the GWSPR sensor, the resonance signal is wider, which is suitable for measuring with angular sensitivity. The so-called GWSPR configuration is the sandwiching of an additional nanodielectric layer between the metal layer and the sensing layer in conventional SPR sensor, and it has been experimentally confirmed by Abdulhalim et al. [25] in 2008. Their results demonstrated that by coating a dielectric thin film (10~15 nm) with a large real part of RI on the surface of a silver based SPR configuration, the sensitivity can be greatly improved due to the enhancement of the electric field intensity at the sensing interface. In this investigation, we proposed a new type of GWSPR configuration, based on an Au–perovskite hybrid structure, to enhance the angular sensitivity of the biosensor.

The perovskite is a general term, representing a structure which can be divided into an inorganic structure [26] and an organic-inorganic hybrid structure [27], which is widely used in solar cells [28,29], photodetectors [30,31], light-emitting diodes [32,33,34], photocatalysis [35,36], lasers [37,38] and other fields. Among these perovskites, CH_3_NH_3_PbBr_3_ [39,40] is a more representative one, which satisfies the requirements as a guided-wave layer (large RI with little absorption) in GWSPR sensors. The complex refractive index of CH_3_NH_3_PbBr_3_ is experimentally measured to be 2 + 0.003i at λ = 633 nm [41]. In the proposed new GWSPR configuration, the CH_3_NH_3_PbBr_3_ is used as a guided-wave layer to form a hybrid structure with an Au film. Our investigation demonstrates that the additional CH_3_NH_3_PbBr_3_ guided-wave layer can greatly improve the electric field intensity of the sensing interface, and thus improve the sensitivity of the proposed biosensor. In the case of the biosensor, the sensitivity of the proposed Au-CH_3_NH_3_PbBr_3_ hybrid structure-based GWSPR biosensor is as high as 278.5°/RIU, which is an improvement of 110.2%, compared with the conventional Au-based SPR sensor. Furthermore, if the Au-CH_3_NH_3_PbBr_3_-based GWSPR configuration is used as a gas sensor, the sensitivity can also be enhanced, by 2.8 times. This investigation shows that perovskite materials not only have an important application value in solar cells, photoelectric detection, and photocatalysis, but also have important application prospects in the field of sensing technology.

## 2. Design Consideration and Numerical Model

The proposed GWSPR biosensor is composed of five components (Figure 1), and these components are prism, Au film, waveguide layer, encapsulation layer and sensing layer. First, the BK7 glass, which has a low refractive index (*n_BK7_*), is used as the coupling prism in the proposed GWSPR biosensor. The *n_BK7_* is defined as the relation [20,42]:(1)$$n_{Bk7}={\left(\frac{1.03961212\lambda^{2}}{\lambda^{2}-0.00600069867}+\frac{0.231792344\lambda^{2}}{\lambda^{2}-0.0200179144}+\frac{1.03961212\lambda^{2}}{\lambda^{2}-103.560653}+1\right)}^{1/2}$$
where *λ* is the wavelength of incident light. In addition to BK7, other prisms such as SF10 and chalcogenide (2S2G: Ge_20_Ga_5_Sb_10_S_65_) have also been calculated and analyzed with the proposed GWSPR configuration. The *n_SF10_* is defined as the following relation [43]:(2)$$n_{\mathrm{SF}10}={\left(\frac{1.62153902\lambda^{2}}{\lambda^{2}-0.0122241457}+\frac{0.256287842\lambda^{2}}{\lambda^{2}-0.0595736775}+\frac{1.64447552\lambda^{2}}{\lambda^{2}-147.468793}+1\right)}^{1/2}$$

The *n_2S2G_* is defined as the relation [44]:(3)$$n_{2S2G}=2.24047+\frac{2.693\times 10^{-2}}{\lambda^{2}}+\frac{8.08\times 10^{-3}}{\lambda^{4}}$$

Second, the Au thin film with a thickness of 50 nm is used as the metal layer to excite the SPR signal, and the refractive index (*n_Au_*) is defined by the Drude–Lorentz model [20,45]:(4)$$n_{Au}={\left(1-\frac{\lambda^{2}\lambda_{C}}{\lambda_{P}^{2}(\lambda_{C}+i\lambda )}\right)}^{1/2}$$
where *λ_C_* = 8.9342 × 10^−6^ m and *λ_P_* = 1.6826 × 10^−7^ m are the plasma and collision wavelengths for Au. Third, the guided-wave layer is CH_3_NH_3_PbBr_3_ perovskite, and its refractive index (*n_per_*) at *λ* = 633 nm is measured to be 2 + 0.003i [41]. Fourth, the cytop material, which has a low reflective index (*n_c_* = 1.34) close to water, is used as the encapsulation layer to protect the GWSPR biosensor, and its thickness is 2 nm. Finally, the sensing medium can be liquid analyte or gas analyte. For liquid analyte, the refractive index is defined as *n_s_* = 1.33 + Δ*n_s_*; for gas analyte, the refractive index is defined as *n_s_* = 1.00 + Δ*n_s_*, where Δ*n_s_* represents some biochemical reaction or concentration change. The change of the biosensing signal can be attributed to the change of the RI. In addition, biological signals can be reflected as changes in plasmon resonance signals, changes in fluorescence signals, and changes in interference signals. In this investigation, a GWSPR signal excited from the Au–perovskite hybrid structure is used to sense the change in the biological signal.

In the multilayer GWSPR structure, the transfer matrix method (TMM) [46] is used for numerical simulation, and the incident light is the He-Ne laser beam (*λ* = 633 nm) in TM-polarized mode. The characteristic transfer matrix (*M*) of the multilayer GWSPR structure is given as:(5)$$M=\prod_{k=2}^{N-1}M_{k}=\left[\begin{array}{l} M_{11}M_{12}\\ M_{21}M_{22} \end{array}\right],\;$$
with
(6)$$M_{k}==\left[\begin{array}{l} \cos\beta_{k}\;\left(-i\sin\beta_{k}\right)/q_{k}\\ -iq_{k}\sin\beta_{k}\;\cos\beta_{k} \end{array}\right]$$
where
(7)$$\beta_{k}=\frac{2\pi d_{k}}{\lambda }{\left(\varepsilon_{k}-n_{1}^{2}\sin^{2}\theta \right)}^{1/2},\;$$
and
(8)$$q_{k}=\frac{{\left(\varepsilon_{k}-n_{1}^{2}\sin^{2}\theta \right)}^{1/2}}{\varepsilon_{k}},\;$$
where *θ* is the incident angle and *n*_1_ is the refractive index of prism (*n_p_*). Then the total reflection coefficient *r_p_* is defined as:(9)$$r_{p}=\frac{\left(M_{11}+M_{12}q_{N}\right)q_{1}-\left(M_{21}+M_{22}q_{N}\right)}{\left(M_{11}+M_{12}q_{N}\right)q_{1}+\left(M_{21}+M_{22}q_{N}\right)}.\;$$

And the reflectance (*R_p_*) is written as:(10)$$R={\left|r_{p}\right|}^{2}.\;$$

Finally, the sensitivity of the GWSPR biosensor is given by [4,47]:(11)$$S=\frac{\Delta \theta }{\Delta n_{s}}.\;$$
where Δ*θ* is the variation of resonance angle caused by the Δ*n_s_*.

## 3. Results and Discussions

The conventional SPR biosensor is shown in Figure 2a, which uses a 50 nm thick Au thin film as the metal layer to excite the SPR. The reflectance curve of the conventional SPR biosensor is calculated by TMM, and it is found that the largest variation of the resonance angle caused by the Δ*n_s_* = 0.002 is Δ*θ* = 0.27°, when the BK7 glass is used as the coupling prism (Figure 2c). Therefore, the sensitivity can be obtained to be S = 135°/RIU. To enhance the sensitivity, another CH_3_NH_3_PbBr_3_ perovskite guided-wave layer is coated on the Au surface to form an Au–perovskite hybrid structure (Figure 2b). When a 10 nm thick CH_3_NH_3_PbBr_3_ perovskite layer is coated on the Au surface, the Δ*θ* will increase to 0.47°, and then the sensitivity is improved significantly to be 235°/RIU (Figure 2d). The enhancement mechanism is that the surface plasmon polaritons can spread along the guided-wave layer by adding a high refractive index perovskite layer. When the guided-wave layer is thin enough, it will result in a large part of the evanescent field being located in the analyte region. Therefore, the detection range is increased, and the sensitivity is improved [4,25]. The electric field distribution of the conventional SPR sensor and the GWSPR sensor are shown in Figure 2e,f for a comparison. The result demonstrates that the electric field at the sensing interface is greatly improved by the additional CH_3_NH_3_PbBr_3_ perovskite layer. Hence, the sensitivity is improved by the guided-wave layer. However, the CH_3_NH_3_PbBr_3_ perovskite is susceptible to pollution and damage, which will reduce the stability and sensitivity of the GWSPR biosensor. Therefore, we coated a cytop material (*d_c_* = 2 nm) as the encapsulation layer on the surface of the guided-wave layer (Figure 1). This encapsulation layer is made of low-refractive index materials with a refractive index close to that of the sensing medium, which can retain the high sensitivity of the GWSPR biosensor and play a protective role.

In the GWSPR biosensor, the thickness of the guided-wave layer is an important factor affecting the GWSPR signal and the sensitivity. When the thickness of the guided-wave layer changes from 8 nm to 15 nm, a significant change in the GWSPR signal can be observed (Figure 3a). Both the minimum R and the resonance angle are changed when there is an increase in *d_p_* (Figure 3b). When *d_p_* ≤ 12 nm, the minimum R (green line) maintains a small range of variation; however, when *d_p_* > 12 nm, the minimum R will change drastically. Similarly, the movement of the resonance angle (red line) is also greatly affected by the *d_p_*. With the increase in the *d_p_*, the resonance angle will move to a larger angle, and the largest resonance angle is 85.45° at *d_p_* = 14 nm. If the *d_p_* continues to increase (*d_p_* ≥ 14 nm), the movement of the resonance angle will be limited by the range of the incident angle (0°–90°). The relationship between the sensitivity and the *d_p_* is presented in Figure 3c. The sensitivity will firstly increase to 278.5°/RIU at *d_p_* = 12 nm, and then decrease rapidly due to the limitation of the angle range. After a calculation, the sensitivity enhancement is shown in Figure 3d, which demonstrates that the sensitivity of the proposed GWSPR biosensor is improved by 110.2%, compared with the conventional SPR sensor.

In the proposed prism-coupling technique-based GWSPR biosensor, the prism is a key component which can have a major impact on sensitivity. The above discussion is based on the BK7 prism. We will analyze and compare the influence of a variety of prisms with different refractive indices on sensitivity. Figure 4a–c shows the GWSPR signals obtained from the prisms of 2S2G, SF10, and BK7. The results demonstrate that the Δ*θ* caused by Δ*n_s_* = 0.002 are 0.082°, 0.174°, and 0.557° for 2S2G, SF10, and BK7 prisms, respectively. The specific sensitivity change under different *n_p_* is shown in Figure 4d, which indicates that a prism with a low refractive index can generate a higher sensitivity than that of a high refractive index. Therefore, the BK7 is used as the coupling prism in the new GWSPR biosensor.

The GWSPR signal will move with the change of *n_s_*, and its change process is similar to the effect of changing *d_p_*. As *n_s_* increases, the resonance signal will move toward a large angle and is limited by the range of incident angle (Figure 5a). However, in the biochemical reaction process, different reaction intensities will lead to changes in Δ*n_s_* (Δ*n_s_* is not a fixed value), which requires the GWSPR sensor to maintain a relatively stable sensitivity under different Δ*n_s_* to ensure the effectiveness of the detection process. Figure 5b shows the sensitivity changes corresponding to different values of Δ*n_s_* at *d_p_* = 12 nm, which indicates that the proposed GWSPR biosensor can maintain a relatively stable sensitivity (S ≈ 278.5°/RIU) when Δ*n_s_* < 0.005, and the corresponding detection range is (1.330, 1.335). In addition, if it is necessary to increase the detection range with stable sensitivity, it can be achieved by adjusting the thickness of *d_p_*. Figure 6a,b show the relationship curves of Δ*θ* and Δ*n_s_* at *d_p_* = 10 nm and *d_p_* = 8 nm, respectively. The results demonstrate that the optimal detection ranges are (1.330, 1.3387) and (1.330, 1.3424) for *d_p_* = 10 nm and *d_p_* = 8 nm, respectively, and the sensitivity can keep in stable values of 243.7 °/RIU (*d_p_* = 10 nm) and 205.6 °/RIU (*d_p_* = 8 nm). The stable sensitivity is inversely proportional to the detection range. In practice, if the detection range needs to be increased, the sensitivity of the GWSPR sensor should be appropriately reduced by changing the thickness *d_p_*.

The proposed GWSPR configuration can also be developed as a gas sensor to detect changes in the concentration of harmful gases in the air, such as NO_2_ [48,49], NH_3_ [50,51], and CO [52,53]. The change of the concentration of harmful gases in the air will lead to the change of the air refractive index. Therefore, the harmful gas content in the air can be analyzed by detecting changes in the refractive index of the air. In the numerical simulation, the change of harmful gas in the air is simplified as the change of air refractive index. The refractive index of pure air is 1, which will change to 1 + Δ*n_s_* when mixed with harmful gases. Herein, the conventional SPR gas sensor is calculated to compare with the new Au-CH_3_NH_3_PbBr_3_-based GWSPR gas sensor. The SPR signal curves of the conventional SPR gas sensor at *n_s_* = 1 and 1.002 are shown in Figure 7a, and the sensitivity is calculated to be 58.5 °/RIU. Under the same Δ*n_s_* caused by the harmful gas, the sensitivity of the proposed GWSPR gas sensor at different thickness of *d_p_* is investigated in Figure 7b. The result shows that the highest sensitivity is 168.5°/RIU at *d_p_* = 41.5 nm, which is 2.8 times higher than the conventional SPR gas sensor. The detailed changes of GWSPR signal modulated by *d_p_* are shown in Figure 7c,d. Moreover, the electric field distributions of the conventional SPR gas sensor and the new Au-CH_3_NH_3_PbBr_3_-based GWSPR gas sensor are calculated for a comparison (Figure 7e,f). The result shows that a significant increase in the electric field at the sensing interface can be observed by coating the guided-wave layer, and then results in the enhancement of the sensitivity.

## 4. Conclusions

In this investigation, a new Au-CH_3_NH_3_PbBr_3_ hybrid structure-based GWSPR configuration is designed to be applied in RI biosensor and gas sensors. First, the CH_3_NH_3_PbBr_3_ thin film is used as a guided-wave layer to coat on the surface of the Au film in the GWSPR sensors, and the results demonstrate that the electric field at the sensing interface can be improved by the additional guided-wave layer. Hence, the sensitivity of the GWSPR biosensor is enhanced to be 278.5°/RIU, which is improved by 110.2% compared with the conventional SPR biosensor. Second, the selection of the prism in the GWSPR biosensor is discussed, and it is found that the prism with a low refractive index can possess a higher sensitivity. Therefore, a low refractive index glass of BK7 is used as a coupling prism in the GWSPR sensor. Third, the designed GWSPR configuration can also be applied in gas sensing, and the highest sensitivity is 168.5°/RIU.

## Figures and Tables

**Figure 1 biosensors-11-00415-f001:**
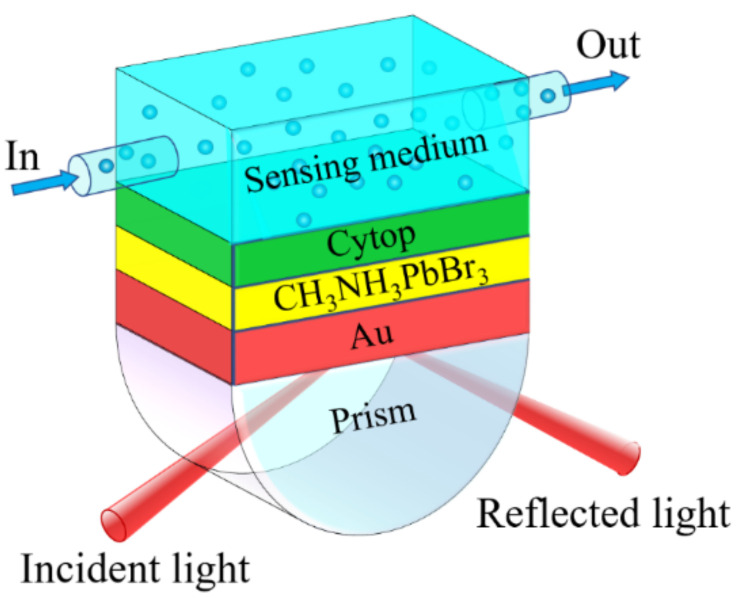
Schematic diagram of the guided-wave surface plasmon resonance (GWSPR) biosensor based on the Au–perovskite hybrid structure.

**Figure 2 biosensors-11-00415-f002:**
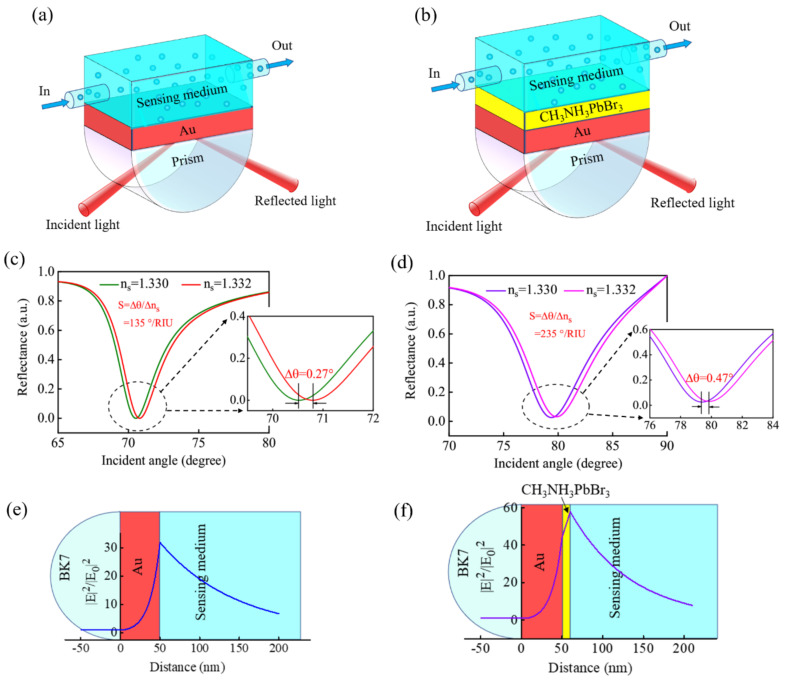
(**a**) Schematic diagram of the conventional Au-based SPR biosensor. (**b**) CH_3_NH_3_PbBr_3_ perovskite as a guided-wave layer to coat on the surface of Au-based SPR biosensor. (**c**) The sensitivity of the conventional Au-based SPR Biosensor at *λ* = 633 nm. (**d**) The sensitivity enhanced by a 10 nm thick CH_3_NH_3_PbBr_3_ perovskite thin film in Au-based SPR biosensor. (**e**) The electric field distribution of the conventional Au-based SPR biosensor. (**f**) The SPR sensor using the guided-wave layer of CH_3_NH_3_PbBr_3_ perovskite thin film to enhance the electric field at the sensing interface.

**Figure 3 biosensors-11-00415-f003:**
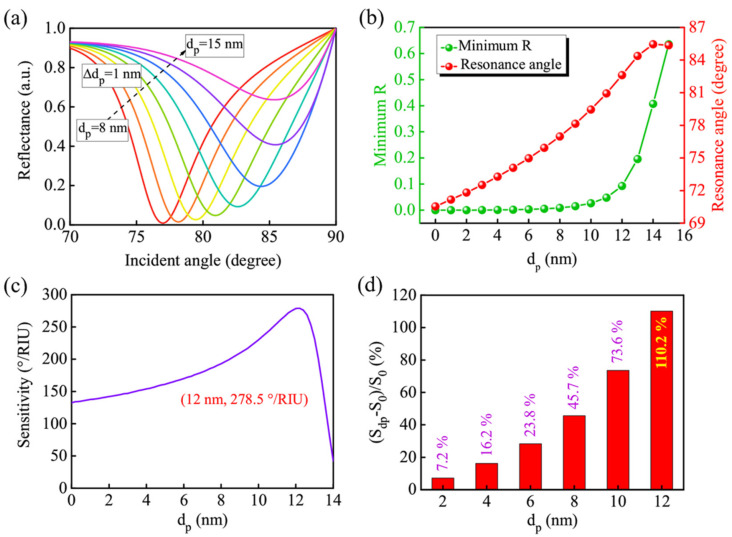
(**a**) The variation of the resonance signal when the thickness of the guided-wave layer changes from 8 nm to 15 nm. (**b**) The variation of minimum R and the resonance angle under different thicknesses of CH_3_NH_3_PbBr_3_ perovskite thin film (*d_p_*). (**c**) The sensitivity of the proposed GWSPR biosensor with different thicknesses of guided-wave layer to obtain the optimal thickness of CH_3_NH_3_PbBr_3_ perovskite thin film. (**d**) Sensitivity enhancement of the proposed GWSPR sensor compared with the conventional SPR sensor. The result demonstrates that the highest sensitivity is 278.5 °/RIU at *d_p_* = 12 nm, which is improved by 110.2% compared with the conventional SPR sensor.

**Figure 4 biosensors-11-00415-f004:**
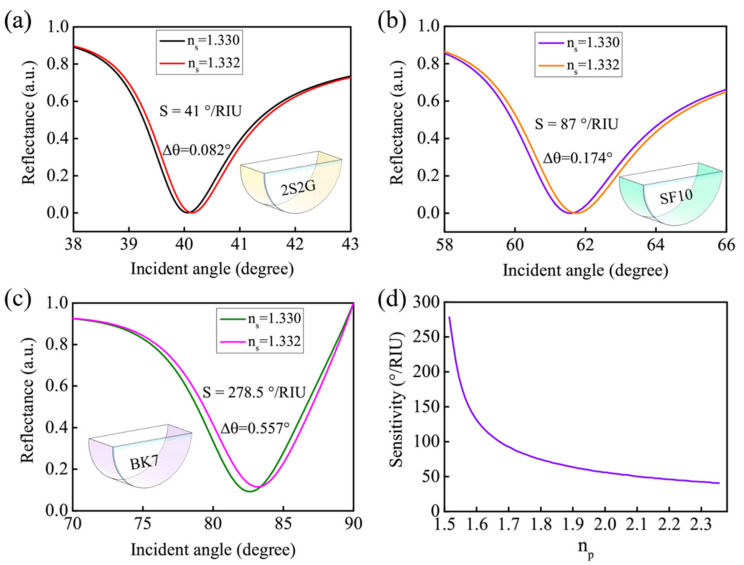
(**a**–**c**) The GWSPR signals excited from the prisms of 2S2G, SF10, and BK7 when Δ*n_s_* = 0.002. The sensitivity is calculated to be 41°/RIU, 87°/RIU, and 278.5°/RIU for the 2S2G, SF10, and BK7 prisms, respectively. (**d**) The specific change of sensitivity with respect to the *n_p_*.

**Figure 5 biosensors-11-00415-f005:**
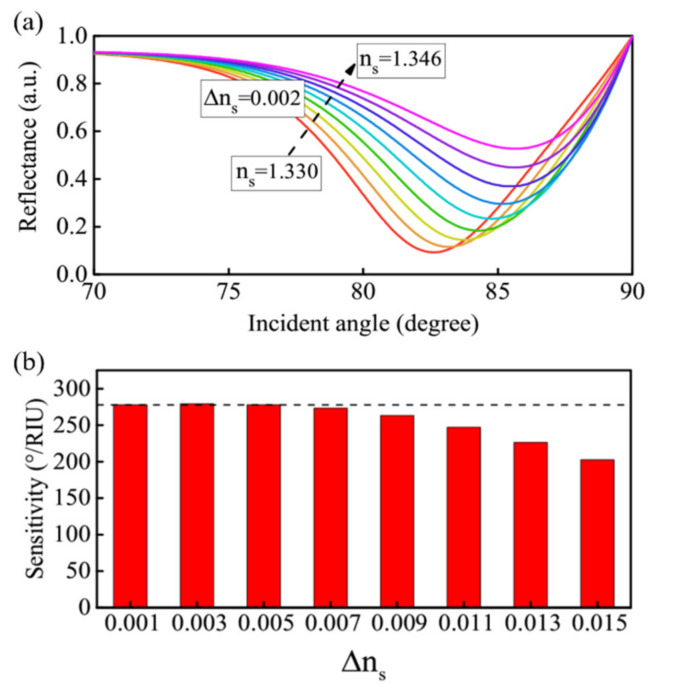
(**a**) The resonance signal obtained from the GWSPR biosensor at *d_p_* = 12 nm when *n_s_* ranges from 1.330 to 1.346. (**b**) The sensitivity obtained from different Δ*n_s_*. When Δ*n_s_* ≤ 0.005, the sensitivity can maintain a stable value.

**Figure 6 biosensors-11-00415-f006:**
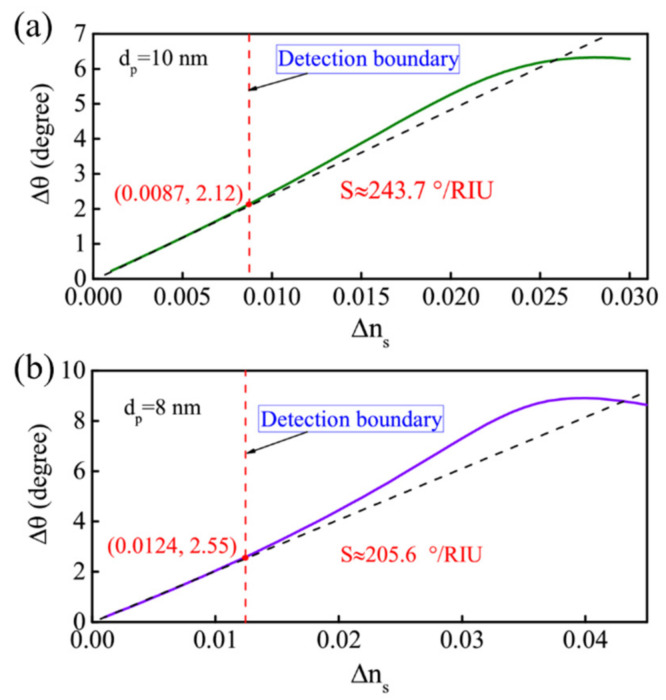
Variation of the Δ*θ* with respect to the different Δ*n_s_* at (**a**) *d_p_* = 10 nm and (**b**) *d_p_* = 8 nm. The results demonstrate that the optimal detection ranges are (1.330, 1.3387) and (1.330, 1.3424) at *d_p_* = 10 nm and *d_p_* = 8 nm, respectively. In these two ranges the sensitivity can keep in stable values of 243.7°/RIU (*d_p_* = 10 nm) and 205.6°/RIU (*d_p_* = 8 nm).

**Figure 7 biosensors-11-00415-f007:**
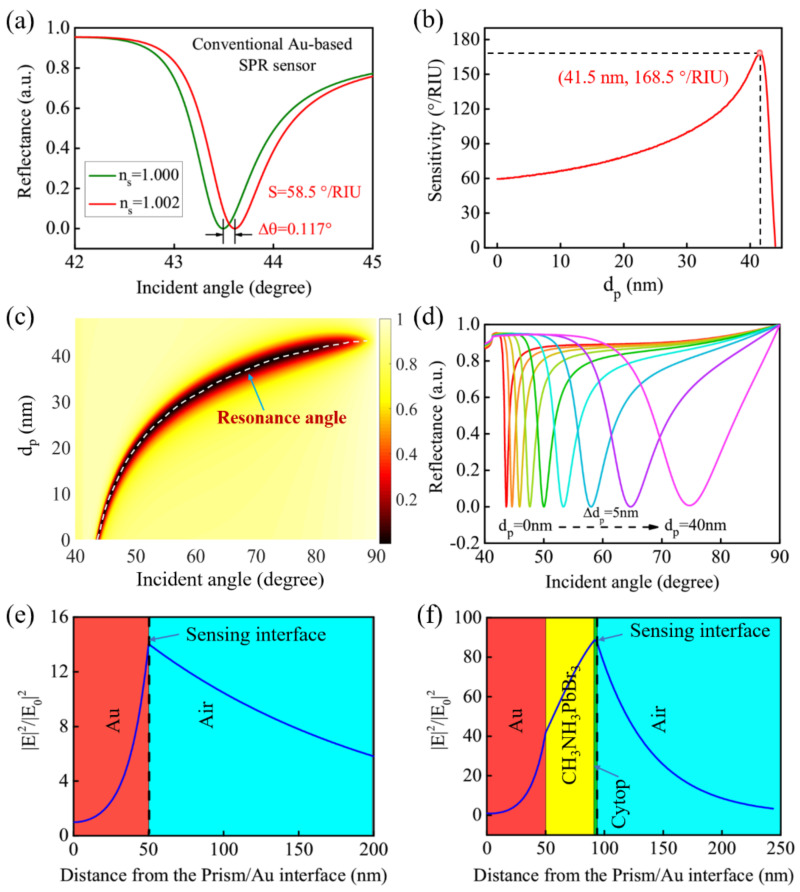
(**a**) SPR signal excited from the conventional SPR gas sensor, and the sensitivity is calculated to be 58.5°/RIU. (**b**) The sensitivity curve corresponding to different thicknesses of *d_p_* for the proposed GWSPR gas sensor, and the highest sensitivity (168.5°/RIU) is obtained at *d_p_* = 41.5 nm. (**c**) The movement of resonance angle corresponding to different thickness of *d_p_*. (**d**) The resonance signals excited from the proposed GWSPR gas sensor at different *d_p_*. (**e**,**f**) The electric field distributions of the conventional SPR gas sensor and the new Au-CH_3_NH_3_PbBr_3_-based GWSPR gas sensor.

## Data Availability

The data presented in this study are available on request from the corresponding author.

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
