# Peer review of "CH3NH3PbBr3 Thin Film Served as Guided-Wave Layer for Enhancing the Angular Sensitivity of Plasmon Biosensor"

_biosensors, 2021, doi:10.3390/bios11110415_

Round 1

Reviewer 1 Report

In this manuscript, the authors theoretically studied the sensing of liquids and gases using a perovskite based guided-wave surface plasmon resonance sensor which shows better performance compared to conventional surface plasmon resonance sensing system based on gold film. The thin film of CH3NH3PbBr3 was coated on the film of Au surface that improves the electric field at the sensing interface, resulting in higher sensitivity. The manuscript is well written and also well organized. Thus, manuscript should be publishable in this journal.  

Author Response

Answer: We thank the referee for the very positive comments on our manuscript.

Reviewer 2 Report

The authors discuss CH3NH3PbBr3 thin film served as guided-wave layer for enhancing the angular sensitivity of plasmon biosensor. While the area is of interest, some changes can be made to improve the quality of the manuscript.

The language of the manuscript sounds vague at some places, for example 

(i) Introduction, first line is too long and needs to be rephrased

(ii) Introduction, first paragraph, last line starting with "However, the conventional SPR..." needs to be rephrased for clarity

(iii) Acronym GWSPR needs to be defined somewhere

(iv) Please rephrase for clarity in Section 2. "
The proposed Au-CH3NH3PbBr3-based GWSPR biosensor consists of five compo- 89 nents (Figure 1), they are prism,..."

Figures 1 and 2 are low in resolution, texts cutoff at places and some images have a fade colour in background.

Are Fig 1 and 2, authors' own figures, if copied from somewhere, source needs to be acknowledged. 

Moreover, have the authors tested sensor performance experimentally in lab or only numerically evaluated. Some experimental evidence or referring to appropriate literature will improve the manuscript.

Author Response

Reviewer #2: The authors discuss CH3NH3PbBr3 thin film served as guided-wave layer for enhancing the angular sensitivity of plasmon biosensor. While the area is of interest, some changes can be made to improve the quality of the manuscript.

Answer: We thank the referee for the very positive comments on our manuscript.

The language of the manuscript sounds vague at some places, for example 

  • Introduction, first line is too long and needs to be rephrased.

Answer: Thanks for your professional comments. The first sentence of the introduction of the original manuscript has been revised into two concise sentences.

  • Introduction, first paragraph, last line starting with "However, the conventional SPR..." needs to be rephrased for clarity.

Answer: Thanks for your professional comments. According to the reviewer’s suggestion, we revised the last sentence of the first paragraph (the section of introduction) to “The conventional SPR biosensor, which is made of 50 nm thick Au film coupled with a prism, has good stability, but the sensitivity is not high enough. Previous reports indicate that the sensitivity of the conventional SPR biosensor based on the BK7 prism is 137 °/RIU [41], which is difficult to detect more slight RI changes of sensing medium. Hence, it is necessary to explore other SPR biosensors with higher sensitivity.”

  • Acronym GWSPR needs to be defined somewhere.

Answer: Thanks for your professional comments. The acronym GWSPR is defined in the second paragraph of the introduction.

  • Please rephrase for clarity in Section 2. "The proposed Au-CH3NH3PbBr3-based GWSPR biosensor consists of five components (Figure 1), they are prism,..."

Answer: Thanks for your professional comments. The sentence in Section 2 is revised to “The proposed GWSPR biosensor is composed of five components (Figure 1), and these components are prism, Au film, waveguide layer, encapsulation layer and sensing layer.”

Figures 1 and 2 are low in resolution, texts cutoff at places and some images have a fade colour in background.

Answer: Thanks for your professional comments. We have re-edited Figures 1 and 2 to improve the resolution of the images.

Are Fig 1 and 2, authors' own figures, if copied from somewhere, source needs to be acknowledged. 

Answer: Thanks for your professional comments. Figures 1 and 2 were designed by us with drawing software and belong to the original Figures.

Moreover, have the authors tested sensor performance experimentally in lab or only numerically evaluated. Some experimental evidence or referring to appropriate literature will improve the manuscript.

Answer: Thanks for your professional comments. At present, the results we show are obtained by numerical calculations, and the relevant experimental verification platform is still under design. In the next research work, we expect to show the relevant sensing results through experiments.

Reviewer 3 Report

The article is written  in a good scientific manner and has its novelty side. From a scientific point of view I can notice the clarity of the expression and the rigor of the application that precedes the experimental part. The authors are invited to highlight the novelty from the literature field. What are the main contributions made by the paper? I recommend also the authors to check  also the bibliography.

Author Response

Reviewer #3: The article is written in a good scientific manner and has its novelty side. From a scientific point of view I can notice the clarity of the expression and the rigor of the application that precedes the experimental part. The authors are invited to highlight the novelty from the literature field. What are the main contributions made by the paper? I recommend also the authors to check also the bibliography.

Answer: We thank the referee for the very positive comments on our manuscript. The abstract has been rearranged to highlight the main contributions of this work. There are three main contributions: 1) It is found that the electric field at the sensing interface is improved by the CH3NH3PbBr3 perovskite thin film, thereby enhancing the sensitivity. The result demonstrates that the angular sensitivity of the Au-perovskite-based GWSPR biosensor is as high as 278.5 °/RIU, which is 110.2% higher than that of conventional Au-based surface plasmon resonance (SPR) biosensor. 2) The selection of coupling prism in the configuration of GWSPR biosensor is also analyzed, and it indicates that a low Refractive index (RI) prism can generate greater sensitivity. Therefore, the low-RI BK7 prism is served as the coupling prism for the proposed GWSPR biosensor. 3) The proposed GWSPR sensing structure can be used not only for liquid sensing, but also for gas sensing, and it is demonstrated that the GWSPR gas sensor is 2.8 times more sensitive than the Au-based SPR gas sensor.

In addition, we re-edited the references of the manuscript to bring it closer to the published version.

Round 2

Reviewer 2 Report

The manuscript looks good in its present form. Authors have addressed the comments from the review.